# Functional Classification and Interaction Selectivity Landscape of the Human SH3 Domain Superfamily

**DOI:** 10.3390/cells13020195

**Published:** 2024-01-20

**Authors:** Neda S. Kazemein Jasemi, Mehrnaz Mehrabipour, Eva Magdalena Estirado, Luc Brunsveld, Radovan Dvorsky, Mohammad R. Ahmadian

**Affiliations:** 1Institute of Biochemistry and Molecular Biology II, Medical Faculty and University Hospital Düsseldorf, Heinrich Heine University Düsseldorf, 40225 Düsseldorf, Germany; neda.jasemi@hhu.de (N.S.K.J.); mehrnaz.mehrabipour@hhu.de (M.M.); radovan.dvorsky@gmail.com (R.D.); 2Laboratory of Chemical Biology, Department of Biomedical Engineering and Institute for Complex Molecular Systems (ICMS), Eindhoven University of Technology, P.O. Box 513, 5600MB Eindhoven, The Netherlands; evamagdalena2@gmail.com (E.M.E.); l.brunsveld@tue.nl (L.B.)

**Keywords:** ARHGAP12, GRB2, NCK1, proline-rich motifs, protein–protein interaction, SH3 domain, signal transduction, SOS1, SRC homology 3, WRCH1/RHOU

## Abstract

SRC homology 3 (SH3) domains are critical interaction modules that orchestrate the assembly of protein complexes involved in diverse biological processes. They facilitate transient protein–protein interactions by selectively interacting with proline-rich motifs (PRMs). A database search revealed 298 SH3 domains in 221 human proteins. Multiple sequence alignment of human SH3 domains is useful for phylogenetic analysis and determination of their selectivity towards PRM-containing peptides (PRPs). However, a more precise functional classification of SH3 domains is achieved by constructing a phylogenetic tree only from PRM-binding residues and using existing SH3 domain–PRP structures and biochemical data to determine the specificity within each of the 10 families for particular PRPs. In addition, the C-terminal proline-rich domain of the RAS activator SOS1 covers 13 of the 14 recognized proline-rich consensus sequence motifs, encompassing differential PRP pattern selectivity among all SH3 families. To evaluate the binding capabilities and affinities, we conducted fluorescence dot blot and polarization experiments using 25 representative SH3 domains and various PRPs derived from SOS1. Our analysis has identified 45 interacting pairs, with binding affinities ranging from 0.2 to 125 micromolar, out of 300 tested and potential new SH3 domain-SOS1 interactions. Furthermore, it establishes a framework to bridge the gap between SH3 and PRP interactions and provides predictive insights into the potential interactions of SH3 domains with PRMs based on sequence specifications. This novel framework has the potential to enhance the understanding of protein networks mediated by SH3 domain–PRM interactions and be utilized as a general approach for other domain–peptide interactions.

## 1. Introduction

Protein–protein interactions are fundamental to the intricate machinery that controls virtually all biological processes. [1]. Among the diverse array of protein domains that facilitate these interactions, the SRC homology 3 (SH3) domains stand out as central modular units. These compact domains, consisting of approximately 60 amino acids with similar sequences that adopt a compact β-barrel fold made of five β-strands [2], are found predominantly in various signaling proteins and various protein families [3,4]. The selective interactions of the SH3 domain with proline-rich motifs (PRMs) are fundamental for the assembly and orchestration of multiprotein complexes [5]. It is rational that the SH3 domain-containing proteins (SH3DCPs) are involved in a wide variety of biological processes [4], subsequently leading to a substantial influence on a spectrum of diseases, such as cancers [6], neurological disorders [7], kidney and urinary disorders [8,9], muscle and myopathy disorders [10,11], immune disorders [12,13], and genetic and developmental disorders [4,14,15].

To date, a series of seven types of PRM-binding modules have been reported, including SH3, WW (two highly conserved tryptophan amino acids), EVH1 (Ena/VASP homology domain 1), GYF (glycine-tyrosine-phenylalanine), Profilin, CAP-Gly (cytoskeleton-associated protein-glycine-rich), and UEV (ubiquitin E2 variant) [16,17]. Proline-rich target peptides possess core PRMs with unique properties that influence interaction selectivity. The diversity of PRMs results from the inclusion of one or more proline residues in various combinations within the peptide sequences. The proline side chains and carbonyl groups are exposed at regular intervals, allowing intermolecular hydrogen bonding with PRM-binding domains [16]. Interactions involving PRMs have a low entropic cost of binding due to the restricted rotational freedom of proline residues along the peptide backbone. This restricted flexibility contributes to a higher overall binding energy for complexes involving PRM-containing peptides (or PRPs). This property increases the affinity of PRM interactions and influences ligand recognition [16]. In addition, PRMs can interact with their binding partners in two distinct orientations; this is influenced by the arrangement of non-proline residues either located N- or C-terminal to the core motif, often involving positively charged counterparts (R or K) [16,18].

In the context of SH3 domain interaction with PRMs, a specific contact recognition occurs, where a positively charged PRM residue binds to conserved negatively charged residues in the variable loop region of the SH3 domain, resulting in moderate selectivity and affinity [5,16]. Comparative analysis of SH3 domain binding sites reveals remarkable variability and flexibility in the loop regions, contributing to the specificity and affinity of PRM binding [6,19]. The preference of the SH3 domain for specific PRMs is usually moderate, with affinities typically in the low micromolar range [20,21,22]. Furthermore, SH3 domains exhibit a broad spectrum of both conventional (PRM-based) and nonconventional selectivity, effectively recognizing a diverse array of protein interactors in a differentiated manner [4,5].

SH3DCPs play a pivotal role in biological processes by facilitating diverse protein–protein interactions that rely on their selectivity and affinity. The intricate nature of protein assembly orchestrated by SH3 domains raises significant questions regarding the underlying selectivity framework governing complex networks of SH3 domain–PRMs interactions [23]. Despite sharing a 25% sequence homology, accurate prediction of the selective PRM recognition by SH3 domains remains a formidable challenge [21,24,25]. In this study, we analyzed the phylogenetic and structure–function relationships of all 298 human SH3 domains, specifically focusing on the sequences of their PRM binding sites. We then performed classification based on their PRM-binding selectivity, organizing them into 10 distinct families. In addition, the distinctive recognition pattern of PRMs within SOS1, a well-established PRM-containing protein, caught our attention. This pattern of selectivity across all SH3 families led us to use SOS1 as a comprehensive model protein to elucidate the recognition mechanisms of established SH3 domains within the human proteome in our research. The binding capabilities and affinities of 25 representative SH3 domains toward 10 SOS1-derived PRPs and 2 reference peptides were carefully evaluated. The reference peptides, RP1, a derivative of SOS1 and part of P3, and RP2, a derivative of the RHO GTPase WRCH1, were used as controls. Our investigation using fluorescence dot blot and polarization techniques revealed a significant finding: out of 300 SH3 domain–peptide combinations, only 45 exhibited binding affinities, which ranged from 0.2 to 125 micromolar. This study pioneers the understanding of the selectivity and affinity of SH3 protein modules for specific PRMs, encompassing a wide range of proteins. This framework lays the foundation for a predictive matrix that enables the anticipation of SH3 domain–PRM-mediated protein–protein interactions within complex cell signaling networks.

## 2. Materials and Methods

### 2.1. Bioinformatics, Databases, and Structural Analysis

The sequences of the SH3DCPs were obtained from the UniProt database by combining full-text searches and sequence homology searches performed with the HMMER v3.4 software package. Isolated sequences of the SH3 domains were then extracted from the previously obtained proteins, again using the subprograms of HMMER. Alignment of the SH3 domain sequences was then performed using BioEdit 7.2.5 software, and the resulting phylogenetic tree was constructed using MEGA 10.2.6 software.

Available structures containing SH3 domains were retrieved from the Protein Data Bank (PDB) website using the BLAST program. To further analyze the SH3 domain structure and define its binding residues for interaction with PRMs, Python scripts were used to identify all residues in SH3 structures within 4.0 Å of the PRM bound to it. The information thus obtained was then projected onto the global sequence alignment of all SH3 domains, and only homologous residues potentially contacting PRMs were selected to define the alignment of the PRM-binding residues of SH3 domains. Finally, PRMs were collected from published articles available on the NCBI website. In addition, BLAST analysis of SOS1 PRMs was performed on the NCBI platform.

### 2.2. Constructs, Peptides, and Proteins

The constructs and peptides employed in our study are listed in Appendix A, respectively. All fluorescein-labeled PRPs were synthesized and used under the conditions described previously [22]. p3XFLAG-CMV ARHGAP12^wt^ [26] was used to generate a SH3 domain deletion (ARHGAP12^ΔSH3^). NCK1^wt^, NCK1^ΔSH3−3^, and NCK1^Set−1^ (N205D, D206T, D226Q, and P227D) were ordered in pcDNA3.0-Flag vectors from BioCat GmbH., Heidelberg, Germany. HA-SOS1 in the pCGN vector was ordered from addgene (#32920). All SH3 domains of the proteins listed in Appendix A within the pGEX4-T1 vector were expressed in *Escherichia coli* strains CodonPlus, Rosetta, and BL21(DE3) and purified as GST-tagged fusion proteins. The purification process involved affinity chromatography on a glutathione Sepharose column [22]. For subsequent polarization analysis, a portion of these GST fusion proteins underwent cleavage of the GST tag using thrombin (#T6884-1KU, Sigma Aldrich, Taufkirchen, Germany) at 4 °C until achieving full digestion of the fusion protein. Following this cleavage step, the proteins were subjected to further purification and separation by employing size exclusion. All purified proteins underwent analysis via SDS-PAGE and were subsequently stored at −80 °C for further use.

### 2.3. Pull-Down and Fluorescence Dot Blot Analysis

Pull-down of 10 µM of FITC-labeled peptides (Appendix A) with 5 µM of purified GST-SH3 domains (Appendix A) was performed using 10 µL of glutathione Sepharose beads (GE Healthcare, Chalfont Saint Giles, UK) in a buffer containing 30 mM Tris-HCl at pH 7.5, 3 mM dithiothreitol, and 5 mM MgCl2 for 1 h at 4 °C. Purified GST was used as a negative control. After three washes, bound proteins were eluted by incubation in the same buffer containing 20 mM reduced glutathione for 15 min at 4 °C, and the beads were separated by centrifugation. Bound FITC-labeled peptides were detected by dot blot analysis using 1 µL of eluent at an emission wavelength of 600 nm and an Odyssey Fc imaging system (LI-COR Biosciences, Lincoln, NE, USA). Detected signals were quantified densitometrically using LI-COR Image Studio version 5.2 imaging software.

### 2.4. Fluorescence Polarization

The interaction between fluorescein-labeled proline-rich peptides (0.2 µM) and increasing concentrations of SH3 domains (ranging from 0 to 200 µM) was measured in a buffer (containing 30 mM Tris/HCl (pH 7.5), 5 mM MgCl_2_, and 3 mM DTT at 25 °C) in a total volume of 200 µL using a Fluoromax 4 fluorimeter in polarization mode and a quartz glass fluorescence cuvette (Hellma Ultra-Micro Cuvette 105.250-QS, Thermo Fisher Scientific, Waltham, MA USA). Excitation was performed at 470 nm and emission was measured at 560 nm. Dissociation constants (K_d_) were determined by fitting the concentration-dependent binding curve to a quadratic ligand binding equation.

### 2.5. Trandfection and Immunoprecipitation Analysis

CHO-K1 cells were cultured in Dulbecco’s modified Eagle’s serum (DMEM, Gibco, Waltham, MA, USA) supplemented with 10% FBS (Gibco) and 1% penicillin/streptomycin (Genaxxon, Ulm, Germany). Cells were co-transfected with HA-tagged SOS1 full-length (FL) and FLAG-tagged NCK1^wt^, NCK1^ΔSH3−3^, and NCK1^Set−1^ or FLAG-tagged ARHGAP12^wt^ ARHGAP12^ΔSH3^ using a Turbofect reagent (Thermo Fisher Scientific). To perform co-immunoprecipitation assays, the CHO-K1 cells were lysed on ice for 5 min employing a buffer containing 50 mM Tris/HCl (pH 7.4), 100 mM NaCl, 2 mM MgCl_2_, 10% glycerol, 20 mM ß-glycerophosphate, 1 mM Na_3_VO_4_, 1% IGPAL (Thermo Fisher Scientific), and 1x protease inhibitor cocktail (Roche, Basel, Switzerland). Lysates were cleared by centrifugation (20,000× *g* at 4 °C for 5 min). Protein concentrations were determined using the Bradford assay. Lysates were then incubated overnight at 4 °C with either anti-Flag M2 agarose beads (Sigma Aldrich) or Protein A-Sepharose beads with anti-Flag antibody and anti-IgG as control. The beads were then washed three times with a wash buffer containing 50 mM Tris-HCl and 150 mM NaCl with 1 mM EDTA. Proteins bound to the beads were eluted with 2.5x Laemmli loading buffer and subjected to SDS-PAGE for further analysis. The primary antibodies used in Western blot analysis included anti-GST (own antibody), anti-Flag (1:1000 WB and 1:50 CO-IP, #F742; and #F3165, both from Sigma), anti-IgG (1:50; # sc-2025, Santa Cruz), anti-SOS1 (1:1000; #sc-256, Santa Cruz), and anti-Vinculin (1:1000; #V9131, Sigma). The secondary antibodies used were purchased from LI-COR (anti-mouse 700 nm: IRDye #926-32213; anti-rabbit 800 nm: IRDye #926-6807).

## 3. Results

### 3.1. Sequence–Structure–Function Classification of Human SH3 Domains

In our previous study, we performed a comprehensive survey in which we identified 298 SH3 domains within 221 SH3DCPs spanning a range of 13- to 720-kilodalton proteins [4]. This analysis included a phylogenetic assessment of human SH3DCPs based on their multidomain architecture, providing a convenient functional classification within different physiological pathways. However, this approach did not address the intrinsic PRM selectivity of the SH3 domain itself. Therefore, we set out to comparatively study the sequence–structure–function relationships of human SH3 domains with a focus on three key aspects: the amino acid sequence, three-dimensional (3D) structure, and spatial arrangement of PRM-binding sites, combining bioinformatics with experimental and structural biology.

As a first step, we focused on elucidating the critical aspects of the PRM binding properties of the SH3 domain. To achieve this, we obtained primary sequences covering a collection of 298 human SH3 domains from the UniProt database. These sequences were then aligned and used to construct a phylogenetic tree of the SH3 domain superfamily using MEGA software (version 10.2.6). The resulting phylogenetic tree, designated tree #1 (Appendix A), depicted the evolutionary relationships among human SH3 domains. Examination of this tree revealed a remarkable conservation of key regions essential for the 3D structure of SH3 domains (Appendix A). This finding underscored a robust and consistent sequence–structure relationship spanning specific parts of the SH3 domain responsible for PRP interactions in SH3DCP families [4]. However, a more complex scenario emerged when delving into the comprehensive analysis of SH3 domain–PRP interactions. Despite meticulous exploration of SH3 domain–PRP structures available in the protein database (Appendix A) and a comprehensive review of published biochemical data on SH3 domain–PRP interactions (Appendix A), no discernible structure–function relationship was revealed from tree #1. Strikingly, PRPs exhibit clustering patterns that are inconsistent with established SH3 domain families. Instead, they were distributed among distantly related families (Appendix A).

A different approach to characterize the SH3 domain–PRM interaction was to perform a second phylogenetic analysis focusing only on the active site regions of the SH3 domain, specifically the residues involved in PRM binding. Unlike the methodology for tree #1, which considered complete SH3 domain sequences, the second phylogenetic tree (tree #2) was constructed exclusively from the PRM-binding residues potentially involved in PRM interactions. The PRM-binding residues within the SH3 domains were inferred from the structures of the SH3 domains in complex with their specific PRPs (Appendix A). It is noteworthy that tree #2 shows 10 different families of SH3 domains that can be very well assigned to their respective PRPs (Figure 1, Appendix A). This strategic approach now allowed us to explore the intricate associations between individual SH3 domains and specific PRMs, as discussed in the following sections.

### 3.2. PRM Selectivities of Different SH3 Families

Classification of PRMs into distinct families based on published structural and biochemical data (Appendix A) reveals specific sequence patterns that guide protein–protein interactions (Appendix A). Family 1 includes motifs such as RX(L/A)PXXP, RXXPXXP, KXX(L/A)PXXP, and PXXP, suggesting a diverse yet structured arrangement for interaction. Family 2, characterized by a PPXPPXP consensus, shows patterns such as XPPX, PXP, PPXPP, PXXP, and PXXXP, indicating a diverse but consistent motif profile. Family 3 sequences, including PXXDY, PXXPXLP, PPPXLP, and PPPPP, show specificity around proline and other residues such as D and Y. Family 4 follows a PXXXPPXPP consensus with specific motifs such as PXXXP and PPXPP. Family 5 has a specific PXPXXP motif. Notably, family 6 lacks structural and biochemical data. However, RIMBP1/2 (RIM-binding protein 1 and 2) can recognize a potential consensus of RXXPXXP and can likely bind to motifs such as the RQLPQL/VP, RLLPPTP, and RQLPQTP found in RIM1/2 (RAB3-interacting molecule 1 and 2). RIMBP1/2 has been shown to bind and couple RIM1/2 to voltage-gated Ca^2+^ channels [27]. Family 7 shows patterns such as PXXPX(K/R), (K/R)XPXXP, (K/R)XXPXXP, PXXPXX(K/R), and PXXP. Family 8 motifs, including PXXXP, PXXXPR, and PXXXPXR, highlight a selective array of proline-rich sequences. Family 9, characterized by PXXPX(K/R) and PXXPX(L/P), shows specificity for proline and amino acid residues such as K/R and L/P. Family 10 presents PX(P/A)XXR, PXXPXXP(K/R), PXXPX(K/R), RXX(K/R)P, and PPPPP motifs, illustrating a specific yet versatile proline-rich arrangement. These results highlight the complex yet diverse nature of PRMs across families controlling specific protein interactions and functions.

We observed an overlap of PRM sequences from families 5 and 6 with family 1, suggesting potential similarities and shared binding motifs within their respective SH3 domain interactions. In our phylogenetic classification based on SH3 domain specificity for PRM, family 1 interacts with RX(L/A)PXXP, RXXPXXP, KXX(L/A)PXXP, and PXXP motifs, whereas family 5 has specificity for PXPXXP and family 6 for RXXPXXP motifs. The biological interpretation of this overlap suggests a potential convergence or similarity in binding preferences among these families despite their specific motifs. Such overlapping PRM sequences imply a nuanced relationship in which different SH3 domain families may exhibit distinct specificities yet recognize certain common motifs. This observation may indicate functional redundancies, cooperative interactions, or shared regulatory pathways among these SH3 domain families in cellular processes.

### 3.3. Affinity and Selectivity of the SH3 Family Proteins for SOS1 PRP

The intriguing recognition pattern of PRMs observed in SOS1, a particularly PRM-rich protein, caught our attention. SOS1 shows co-occurrence of 13 out of a total of 14 PRMs (including sequences such as PPPP, XPPX, PXP, PXPXP, PPXPP, PXXP, PXXPX[KR], [KR]XXPXXP, PXXPXXP, PXXXP, PXXXPXXXP, PXXXPR, and PXXXXP), as shown in Appendix A. This distinct pattern, showing selectivity across all SH3 families, motivated us to use SOS1 as a comprehensive model to uncover the recognition mechanisms of established SH3 proteins in the human proteome. Therefore, 25 SH3 domains from different SH3 families were selected (Figure 1), cloned, purified as GST fusion proteins, and used for PRP binding analysis (Appendix A). We selected at least one representative SH3 domain per defined family concerning accessibility and experimental viability (see Appendix A). In addition, we selected 10 different PRPs from the proline-rich domain (RPD) of the SOS1 protein (Appendix A). This collection was designed to cover the full spectrum of PRM types (P1-P10; Appendix A). Two reference peptides were included as controls: RP1, a well-studied SOS1 derivative encompassing part of P3, and RP2, a peptide derived from the N-terminal extension of the RHO GTPase WRCH1/RHOU (Appendix A). The 12 PRPs were labeled with FITC to assess their binding capacities with purified GST fusion proteins of the 25 SH3 domains using fluorescence dot blot and polarization analysis (Figure 2A).

The binding of 12 FITC-labeled PRPs to 25 GST-SH3 domains was qualitatively analyzed by combining GST pull-down and dot blot assays. GST protein alone was used as a negative control. In a previous study [22], we showed that fluorescein labeling does not affect the interaction of proline-rich peptides with the SH3 domains. Different binding strengths were observed among the proteins tested, particularly with the P2, P3, P4, P7, and P9 peptides (Figure 2A). The strongest interactions (dot intensity >80) were between ABI1 and P2, ITSN1-1 and RP2, and NCK1-3 and P9 and RP1. In contrast, no PRP binding (dot intensity 0) was detected for NCK1-1, NPHP1, RASA1, SH3GLB-1, SNX9, ITSNS1-2, ITSNS1-3, and ITSNS1-4. These data provide valuable insight into the varying degrees of interaction across the panel of PRPs and SH3 domains tested (Figure 2A).

Fluorescence polarization measurements were performed to determine the binding affinities of SH3 domain–PRP interaction pairs from the dot blot analysis. SH3 proteins were titrated at increasing concentrations against fluorescent PRPs, which were kept at a constant concentration of 0.2 µM. GRB2-2^W193K^, which is defective in the binding of PRPs such as RP1, was used as a negative control as previously described [22]. Interestingly, none of the PRPs we examined showed any binding for seven SH3 domains: ITSN1-2/-3/-4, NPHP1, RASA1, SH3GLB1, and SNX9 (Appendix A). The resulting data (Appendix A) allowed the evaluation of equilibrium dissociation constants (K_d_) for 45 interactions between the SH3 domains and the PRPs (Figure 2B; Appendix A). In particular, the results confirmed that the peptides P2, P3, P4, P7, and P9 were associated with approximately 17 SH3 domains. The K_d_ values determined were categorized into four affinity levels (Figure 2B; Appendix A): high (0.1 to 1.0 µM; green), intermediate (1.1 to 5 µM; blue), low (5.1 to 25 µM; red), and very low (26 to 125 µM; black). Whereas previously reported SH3 domain–PRM interactions exhibited micromolar affinities, our results revealed interactions with nanomolar affinities in some cases. The most notable and novel pairs of interaction were ARHGAP12/P7, NCK1-3/P9, and NCK1-2/RP2, which had affinities in the nanomolar range of 0.2, 09, and 1.0 µM, respectively.

### 3.4. Non-Conserved Residues Define the Selectivity and Affinity of SH3 Domain–PRM Interactions

To better understand the role of the residues of SH3 domains in selective binding to PRPs, we have generated a multiple sequence alignment (Appendix A). It highlights the conserved and variable residues that are likely to be critical for the selectivity and affinity of the SH3 domain–PRM interactions. The importance of variable residues was investigated by specifically selecting ARHGAP12 and NCK1-3 for mutational analysis due to their high binding affinities of 0.2 and 0.9 µM for P7 and P9, respectively. In contrast, ABL2 and BIN1 were selected for their very low affinity for P7 and P9, respectively. Two different sets of mutations were generated by substituting a combination of amino acids from ARHGAP12 and NCK1-3 for ABL2 and BIN1 and vice versa (Figure 3A; Appendix A).

Comparative fluorescence polarization measurements between wt and mutant SH3 domains (Appendix A) revealed that variable residues determine selectivity and affinity. The determined K_d_ values of 14 and 23 µM showed a drastic reduction in the binding affinity of the Set-1 mutants of ARHGAP12 and NCK1 by 115-fold and 15.6-fold for P7 and P9, respectively (Figure 3B). This demonstrates the importance of the selected variable residues for the PRP interactions, especially because ABL2^Set−1^ and BIN1^Set−1^ showed a 9- and an 11-fold increase in binding to P7 and P9, respectively (Figure 3B). In light of this result, we decided to investigate another set of variable residues (Set-2; Figure 3A). The binding affinity for ARHGAP12^Set−2^ was reduced 47-fold, indicating the critical role of these residues in determining the selectivity and affinity of ARHGAP12 for P7 (Figure 3B). However, NCK1-3^Set−2^ did not differ from the NCK1-3^wt^ in terms of P9 binding, suggesting that the Set-2 residues are not critical for NCK1-3/P9 interaction (Figure 3B).

To investigate a potential SOS1 binding of ARHGAP12 and NCK1 and to assess the relevance of variable residues in the SH3 domain–PRMs interaction in cells, CHO-K1 cells were co-transfected with wt and mutant variants of NCK1 and ARHGAP12 together with HA-SOS1 containing P7 and P9 at its C-terminal PRD (Figure 4). Co-immunoprecipitation (Co-IP) with anti-Flag beads was performed to investigate the possible interaction of ARHGAP12 and NCK1 with HA-SOS1 (Appendix A). As shown in Figure 3C, all three NCK1 proteins, wt, ΔSH3-3, and Set-1, were immunoprecipitated, but SOS1 was only co-immunoprecipitated with NCK1^wt^ and not with NCK1^ΔSH3−3^ and NCK1^Set−1^ (Figure 3C). No ARHGA12-SOS1 interaction was observed in similar experiments (Appendix A). Taken together, our data not only highlight the essential role of these flanking residues in determining the selectivity and affinity of SH3 domains for their cognate PRPs but also provide unprecedented insight into a potential SOS1-NCK1 interaction in cells. 

### 3.5. SH3 Domain–PRP Relationships beyond SOS1

We used the position-specific iterated BLAST (PSI-BLAST) algorithm to perform an analysis of the SOS1 PRPs in the human proteome using the NCBI (National Center for Biotechnology Information, Bethesda, MD, USA) database as a reference. Our goal was to identify homologous peptides from proteins other than SOS1 as potential interaction partners for the SH3 domains investigated in this study. In our analysis, we considered alignments with percentage identities ranging from 98% to 100% and with E values up to 10, which, although less stringent, may still indicate potential similarities between the SOS1 PRPs and other protein sequences. More than 30 proteins were found with partial sequence identities with P2 to P9 (Appendix A; Figure 4 and Appendix A) and 0 with P1 and P10 (Refer to Appendix A). Some of these proteins contain multiple PRM repeats (Appendix A), for example, IQSEC2, paxillin, DLGAP1, PI3KAP1, and WRCH1/RHOU within P3 (Motif: PVPPPVP), SSTR5 and SLX4 within P7 (Motif: PPPPQTP), DCAF1 and MAGED4 within P8 (Motif: HLPSPP), and SOS2 and HCG2013210 within P9 (Motif: PPVPPRQ). This suggests that the SH3 interactions with the PRPs characterized in this study go beyond SOS1 as a binding partner, although the binding specificities of the listed proteins (Figure 4) remain to be investigated. However, several studies confirm the interactions of the identified proteins with SH3DCPs, including zinc finger proteins with p130Cas [28], MACF1 with Spectrin [29], DLGAP with DLG [30], WRCH1/RHOU with GRB2 [31,32], Paxillin with SRC [33,34], and SSTR5 with Homer, Dynamin, IRSp53, and Cortactin [35].

## 4. Discussion

SH3 domains are critical in multiple signaling pathways; they interact with diverse proteins involved in apoptosis, proteasomal degradation, endocytosis, and with SRC family protein tyrosine kinases, influencing downstream processes including proliferation, cell survival, growth, actin reorganization, and cell migration [4]. The broad influence of SH3 domains on cellular functions raises fundamental questions about the specificity of their interaction networks. Previous research has also highlighted the importance of proline amino acids in forming the polyproline type II helix (PPII) conformation, which provides a binding pocket for SH3 domain residues, particularly from the RT and n-SRC loops [17]. Despite the discovery of 14 PRM consensus sequences in the human proteome (Appendix A), the specificity pattern of the interaction of SH3 domains with PRMs is still unclear. Understanding the molecular basis of SH3 domain–PRM interactions is crucial to gain insight into how these interactions regulate signaling pathways.

Interface residues often play an important role in the functional outcome of protein interactions. Many diseases, including cancer and neurodegenerative disorders, are associated with aberrant protein–protein interactions resulting from mutated interacting residues. Abnormal SH3 domain interactions in cancer fuel dysregulated signaling that drives uncontrolled cell proliferation, survival, tumor growth, metastasis, apoptosis evasion, and resistance to anticancer therapies [4,6]. In neurodegenerative diseases like Alzheimer’s and Parkinson’s, these interactions disrupt signaling pathways, leading to the accumulation of misfolded proteins and neuronal degeneration. Altered SH3 domain interactions in synaptic proteins affect neurotransmission and synaptic plasticity [7]. In autoimmune diseases, aberrant SH3 domain interactions contribute to immune cell activation and tissue damage, as immune cells mistakenly target healthy tissues [12,13]. Understanding the interface residues involved in SH3 domain–PRM interactions can thus provide insight into disease mechanisms and potential targets for intervention. In drug development, knowledge of interface residues is essential for designing molecules that can disrupt or modulate specific protein–protein interactions. Targeting these residues can lead to the development of therapeutic agents for the treatment of various diseases.

In recent years, several studies have been devoted to elucidating the diverse nature of SH3 domain interactions. Cesareni and colleagues used a novel chip technology to perform high-throughput qualitative analyses, revealing a variety of human SH3 domains that fall into two categories: those characterized by classical proline-rich core motifs accompanied by positively charged amino acids and atypical ones lacking the core motif [20]. Nevertheless, the diversity of PRM selectivity patterns is evident among all human SH3 domains. Furthermore, a comprehensive analysis of SH3 domain interactions in the evolution of four yeast species, Saccharomyces cerevisiae, *Ashbya gossypii*, Candida albicans, and *Schizosaccharomyces*, revealed that nearly 75 percent of SH3 families identified within the phylogenetic tree have a conserved SH3 specificity profile over 400 million years of evolution [36]. Utilizing the evolutionary relationships of peptide recognition domains in eukaryotes, we identified common structural features and ancestry that allowed us to group SH3 domains into similar binding preference families. This comprehensive investigation aimed to clarify the specificity profiles of SH3 PRMs within the human proteome through categorization based on the phylogenetic tree of SH3DCPs.

To provide an accurate specificity map of SH3 domains, we performed deep phylogenetic analyses coupled with computational analysis of the related structural data. Initial evolutionary analysis of the sequence–structure–function of full-length SH3 domains was unsuccessful due to the presence of SH3 regions that do not interact with PRMs. Within these, each SH3 domain exhibited variation in binding specificity to PRMs, making the characterization of SH3DCPs infeasible. Instead of relying on full-length sequences, we focused on binding residues that directly interact with PRMs, as revealed by sequence alignments coupled with analysis of the published SH3 domain structures in complex with PRMs. This refined phylogenetic approach led to the identification of ten distinct families based on both the structural and biochemical assessments of SH3 domain–PRM interactions and their distribution within the phylogenetic tree. This approach facilitates the assessment of cross-reactivity among SH3 domain recognition sites for PRMs, a phenomenon also observed in previous studies examining the yeast SH3 domain peptide library [37] and including SH3 domains that recognize multiple PRMs. The findings of this study highlight the fact that each SH3 domain family interacts with different but distinct sets of PRMs.

Considering the significant involvement of SOS1 in interactions with SH3DCPs such as GRB2 [38], ITSN1 [39], NCK1 [40], and ABI1 [41], along with its comprehensive coverage of all known PRMs in the human proteome (Appendix A), we decided to use SOS1 as a model for in-depth exploration of SH3 domain specificity in the realm of polyproline interactions. We performed in vitro studies with 25 representative SH3 domains selected from the phylogenetic tree. We performed low-throughput analyses, including pull-down assays, dot blotting, and fluorescence polarization, to investigate SH3 domain–PRP interactions. These investigations revealed novel interactions that had nanomolar affinities, which were subsequently confirmed by mutational studies.

The general concept of protein association is essential for describing protein–protein interactions in complexes, especially those with weak affinities in the micromolar range or transient interactions such as the SH3 domain with PRMs [42]. Mayer and Saksela noted that the limited selectivity of SH3 domains for PRMs implies that SH3-domain-mediated interactions may be highly dependent on external environmental factors [43]. In certain scenarios, the presence of additional surfaces on either the SH3 domain or the ligand it recognizes, along with the presence of either multiple SH3 domains or different domains within the same protein, or even the co-localization of two partners within a multi-protein complex, can cooperatively enhance SH3 domain–PRM specificity to a significant degree [44]. This suggests that the low-affinity-region results in our study may be compensated by these scenarios, ultimately increasing the affinity and specificity of SH3 domain–PRM interactions. It has also been reported that the moderate affinities of SH3-domain-mediated interactions imply that the interactions have a high dynamic remodeling potential (rapid off-rates), depending on the subcellular localization and accessible binding partners [43]. This observation is consistent with our polarization data, which showed high K_d_ values for many low-affinity interactions.

The question of how the specificity of SH3 domain–RPM networks is achieved has been addressed by various research groups. It has been postulated that specificity in cells is not solely encoded by isolated SH3 domain–PRM partners but rather by the context in which the partners are presented as full-length proteins. Dione et al. have shown that the identity of the host protein and the position of the SH3 domains within their host are critical for interaction specificity, cellular functions, and key biophysical processes such as phase separation [45]. In addition, Zarrinpar et al. have shown that isolated SH3 domains can determine the interaction specificity between host SH3 domains [46]. This may also be true for certain high-affinity SH3 domain–PRM interactions, as shown in this study for the newly discovered interactions of NCK1-2, NCK1-3, and ARHGAP12 with WRCH1/RHOU-derived RP2 and SOS1-derived P9 and P7, respectively. NCK1 has been shown to modulate ITSN1-CDC42-WASP-dependent actin polymerization [47]. WRCH1/RHOU, a CDC42 homologous protein, encompasses an extended N-terminus that contains RPMs specific for various adaptor proteins, including GRB2, CRK, and NCK1 [32]. The association of these proteins with WRCH1/RHOU may not only determine its signaling specificity but may also regulate its activity in cells [48]. However, it should be noted that, in some cases, a negative effect of other domains on SH3 domain–PRM binding was observed. Notably, NCK1-3 showed more extensive protein interaction than the full-length NCK1 in immunoprecipitation experiments, possibly indicating a detrimental effect of the C-terminal SH2 domain on specific SH3-domain-mediated interactions. Furthermore, the spectrum of proteins associated with NCK1-3 is not simply the cumulative sum of proteins associated with individual SH3 domains [49]. 

A closer examination of the PRMs revealed that the positioning of the proline residues plays a critical role in the recognition of the SH3 domain, providing the structural basis for defining interaction specificity. The current results indicate that residues −2, −1, +1, and +2 are critical for the recognition of SH3 PRMs. In addition, adjacent positively charged residues contribute additional features that help stabilize the transient interaction [50]. For example, structural studies of the PI3K SH3 domain in association with the peptide RKLPPRPSK provided evidence for the role of adjacent non-proline residues such as Arg-1, Leu-3, and Arg-6 in contributing to the SH3 domain interaction [51]. Arginine residues at positions R + 5, R + 6, and R + 7 are thought to play an important role in enhancing the affinity of GRB2-SH3 domains for SOS1-PRM by contributing to the overall free energy of the interaction [52]. While the majority of SH3 domains interact with PRMs, there have been documented cases where the SH3 domain deviates from the typical classical proline-rich interaction pattern [4]. For example, the RASA1 (or p120RASGAP) SH3 domain specifically interacts with the catalytic arginine finger of the RHOGAP domain of DLC1, thereby competitively and potently inhibiting its RHOGAP activity [53]. Interestingly, none of the PRPs we examined showed any binding for seven SH3 domains: ITSN1-2/-3/-4, NPHP1, RASA1, and SH3GLB1 (Appendix A). While our study comprehensively highlights the major SH3 domain–PRM interactions in the human proteome, the specificity and mechanism of the PRM-independent interaction of SH3 domains remain to be elucidated in further studies.

In our study, SOS1 was used as a PRP model for biophysical and bioinformatic analysis of the SH3 domain–PRM interaction landscape because the SOS1 PRD contains 13 out of 14 different classified proline-rich consensus sequence motifs (Appendix A). This alternative model reveals a spectrum of interactions between different SOS1 PRPs and a number of SH3DCPs, including ABI1, ABL2, ARHGAP12, ARHGEF30 (Obscurin/OBSCN), BIN1, CRK-1, DLG2, GRB2, ITSN1, NCK1, SRC, SH3PXD2A-1, and SORBS1-1. Among them, ABI1 [54], ITSN1 [55], SRC [56], NCK1 [57], GRB2 [58], CRK [59], and SH3PXD2A (TKS5) [60] have been previously established as SOS1 binding partners in cells. ABL1, but not ABL2, has also been shown to interact with SOS1 [61]. Importantly, the precise binding sites for most of these proteins have yet to be investigated. Our study not only elucidates the binding sites of these established SOS1 partners but also uncovers novel interactions, including ABL2, BIN1, DLG2, SORBS1, ARHGEF30, and ARHGAP12. In particular, a high affinity (<0.5 µM) interaction was observed between ARHGAP12 and the P7 of SOS1, demonstrating the interplay between small GTPase regulators, GAPs, and GEFs. However, immunoprecipitation experiments with overexpressed ARHGAP12 in CHO-K1 cells did not confirm an interaction with SOS1 (Appendix A). The reliability of this result may depend on the expression level and affinity of other interaction partners, like accessory proteins, that could potentially bind more strongly and possibly in a multivalent manner. However, it is important to note that this result does not definitively rule out the existence of this interaction in cells, especially considering examples of reported GEF-GAP interactions. A study using immunoprecipitation and mass spectrometry unravels the intricate protein interaction networks involving the synaptic proteins SYNGAP1 (RASGAP), KALIRIN (RHOGEF), and AGAP2 (ARFGAP) in both the postsynaptic density (PSD) and non-PSD fractions of the adult mouse cortex. This investigation sheds light on their role in the organization of GAP and GEF protein families and their associations with proteins associated with intellectual disability and psychiatric disorders [62]. In conclusion, to confirm the significance and broader implications of these novel findings, additional studies within the cellular context are warranted.

Predicting the potential interaction of SH3 domains with PRMs by considering their sequence specificities, as we did in this study, is a promising approach in the field of molecular biology and protein–protein interactions. In pursuing this goal, we are faced with an interesting challenge: the identification of binding affinities between SH3 domains and peptides containing PRMs. The sequence specificity of these interactions is paramount, as SH3 domains exhibit diverse binding preferences that depend on the PRMs present in the peptides.

## 5. Conclusions

SH3 domains are small protein interaction modules that are involved in numerous fundamental cellular processes and associated with the development of several diseases, including Joubert syndrome, leukemia, lymphoma, Usher syndrome or non-syndromic deafness, centronuclear myopathy, schizophrenia, and other neurodevelopmental disorders [4]. Over the past three decades, researchers have focused on how members of the SH3DCP superfamily selectively recognize and bind to their associated PRM-containing proteins.

To systematically address this question, we first extracted 298 SH3 domains in 221 SH3DCPs ranging in size from 13 (small monodomain proteins) to 720 (large multidomain proteins) kilodaltons [4]. The subsequent evolutionary multidomain relationship of the SH3DCP superfamily not only allowed us to functionally classify them into thirteen families but also provided new insights into their diverse roles and interactions in cellular signaling processes, as well as their relevance to various diseases when exploiting the modular interactions of SH3 proteins as drug targets [4]. In the present study, we have incorporated the available sequence, structure, and interaction data into a phylogenetic tree (Figure 1) that groups 298 SH3 domains in the human proteome into 10 families related to the PRM binding interface. These families are aligned with the frames necessary for the interaction of their respective potential PRMs. Mutational analysis suggests the critical role of non-conserved sequences within each SH3 family in defining the specificity and affinity of their interactions with specific PRMs. This investigation highlights that the recognition mechanisms of SH3 proteins across the human proteome are not only influenced by PRMs but also by the core binding site within the SH3 domain. The study of the PRM-binding residues of SH3 domains revealed a significant relationship between individual SH3 domains and specific PRMs, culminating in a detailed map of their associations.

In this study, we performed a comprehensive analysis by comparing published biochemical interaction data, available structural information, and sequence alignments. The goal was to identify specificity-determining residues within PRMs that are critical for interacting with different SH3 domains. A phylogenetic tree based solely on the interacting interface of SH3 domains allowed us to categorize distinct families of SH3 domains within the human proteome, each interacting specifically with unique PRMs. Subsequent mutational analysis supported our categorization and hypothesis by demonstrating that the non-conserved interface sequences within each family are critical in defining the specificity of their interaction with PRMs. The different interface residues within each family were found to determine the affinity and specificity of each protein towards PRMs. In particular, the discovery of common PRMs in two different SH3 domain families underscores the importance of other residues (designated X) beyond proline in determining interaction specificity. It is generally accepted that prolines serve as recognition sites and the backbone of interactions, while X residues define specificities. A comparison of the PRM consensus sequences of the SOS1-homologous proteins reveals other common amino acids, such as V in P3 and P9, Q in P7 and P9, R in P3, P4, P5, and P9 (Appendix A). Notably, the third SH3 domain of NCK1 (NCK1-3) tightly bound P9 but none of the other PRPs tested in this study (Appendix A); this implies that residues other than valine and arginine in P9 may dictate the specificity of the NCK1–P9 interaction. In addition, the sequence motif ^224^ENDPEW of NCK1-3 (Figure 3A) contains three negatively charged residues that may be in electrostatic contact with the R in P9. In contrast, NCK1-1 and NCK1-2, which do not bind P9, contain a lysine instead of glutamate or aspartate within this sequence motif, which counteracts a P9 interaction. An important next step in elucidating the specificity of the interaction of the SH3 domain with the PRM at the atomic level is to analyze the nearly 800 available experimental structures containing SH3 domains. This will be performed by generating homology models and correlating and combining them with the measured affinities and known binding properties of SH3 domain–PRM complexes.

A total of 7 out of the 25 examined SH3 domains showed no interaction with any of the 12 selected PRPs (Appendix A). The tested peptides cover 13 of the 14 recognized proline-rich consensus sequence motifs, suggesting that they may bind in a proline-independent manner. (Appendix A). It is suggested that these SH3 domains may bind in a proline-independent manner. SH3 domains in several studies exhibit an extended repertoire of binding sequences, known as proline-independent binding, allowing SH3DCPs to mediate a broader array of interactions [4,5]. An example of atypical binding is the SH3 domain of RASA1, the RAS-specific GAP (p120RASGAP), which interacts with the catalytic GAP and kinase domains of DLC1, thereby inhibiting its activity [53,63]. Another example is the selective interaction of the GADS/GRAB2L SH3 domain with an RXXK motif of SLP-76 [64]. These types of SH3 interactions with a non-canonical binding mode add to the complexity of understanding protein–protein interactions involving SH3 domains.

The SH3 domains are modular building blocks across all five kingdoms of life and viruses and play a critical role in facilitating inter- and intramolecular interactions and functional interplay within domain-specific interaction networks. SH3DCPs, except the MIA family with a single SH3 domain, are multi-domain proteins [4]. Several recent studies have shown that SH3 domains have an extended repertoire of binding sequences, known as proline-independent binding [4,17,53]. This allows SH3DCPs to mediate a wider range of interactions. However, a quantitative description of the communication between two different sites in a multivalent protein is still challenging. In some cases, the task reaches another level of complexity, such as the interaction of the two SH3 domains of GRB2 with SOS1. Not only the association of the two functional SH3 domains of GRB2 with SOS1 but also the physical interactions between the two SH3 domains are required to allosterically control SOS1 activation [22]. Such a regulatory mechanism involves a series of intramolecular interactions that are further amplified by the interaction of GRB2 with upstream ligands [65].

## Figures and Tables

**Figure 1 cells-13-00195-f001:**
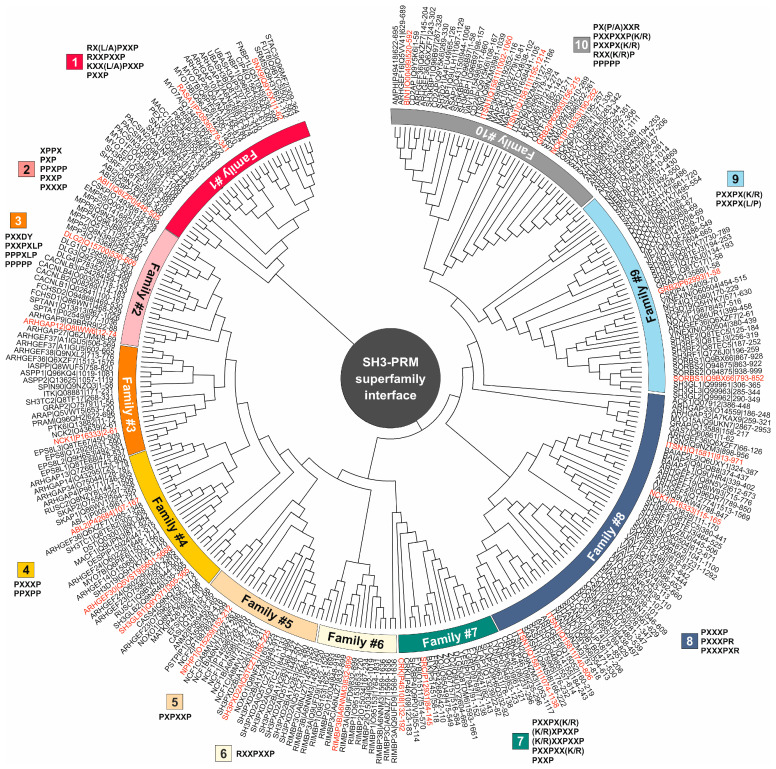
Evolutionary relationships among PRM-binding residues within SH3 domains. The phylogenetic tree (tree #2) was constructed by collecting PRM–-binding residues from 298 human SH3 domains, using published SH3 domain-PRM structures (Appendix A) and biochemical data (Appendix A), and utilizing the MEGA7 software. SH3 domains were systematically classified into ten distinct families based on their interaction properties with specific PRMs as indicated by the color codes. SH3 domains highlighted in red were selected as representatives for further analysis in this study. The protein name is accompanied by the corresponding Uniprot ID and residue span for the SH3 domain.

**Figure 2 cells-13-00195-f002:**
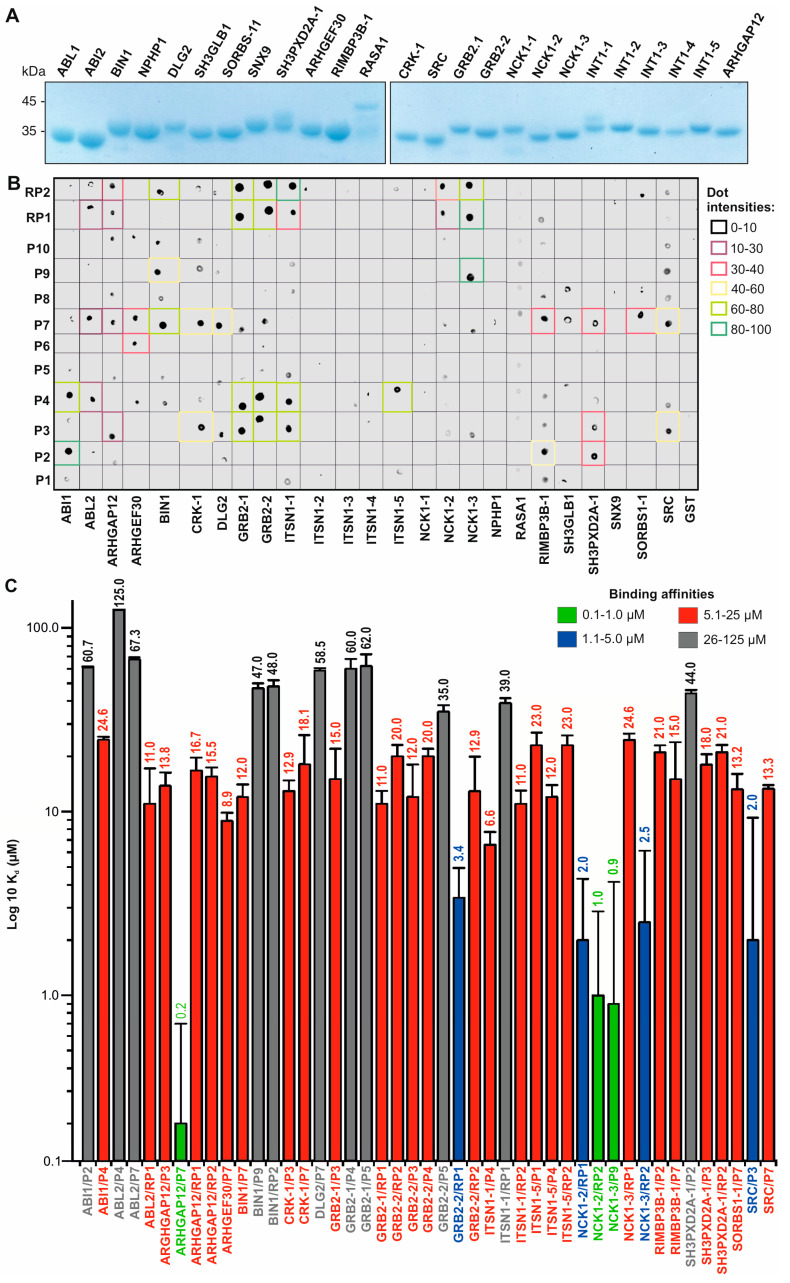
Evaluation of the binding selectivity of SH3 domain representatives with different PRP types. (**A**) Coomassie brilliant blue-stained SDS gels show purified SH3 domains as GST fusion proteins. (**B**) Fluorescence dot blots revealed the variable binding strengths of 12 fluorescent PRPs with 25 GST-SH3 domains. Dot intensities are categorized into five groups ranging from 0 (black) to 100 (dark green). (**C**) Bar graphs show the evaluated dissociation constants (K_d_) for the selected SH3 domain–PRP interactions determined by fluorescence polarization (Appendix A). The color codes indicate the K_d_ values, classified into high affinity (green), intermediate affinity (blue), low affinity (red), and very low affinity (black).

**Figure 3 cells-13-00195-f003:**
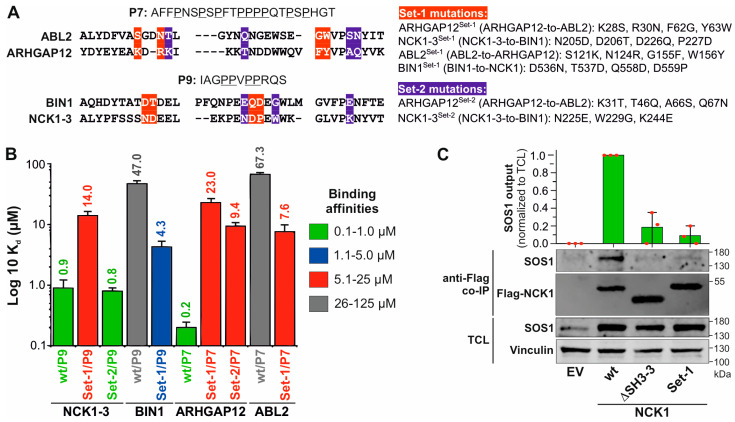
Variable residues determine the specificity and affinity of SH3 domain–PRP interactions. (**A**) A sequence alignment of the PRM-binding residues of ABL2, ARHGAP12, BIN1, and NCK1-3 is extracted from the alignment shown in Appendix A. Residues in red (Set-1) and blue (Set-2) are variable residues (left panel) and are the subject of mutation analysis (right panel). (**B**) K_d_ values determined by fluorescence polarization partially revealed shifts in the binding affinities of the investigated SH3 domain mutants for P7 and P9, respectively, with decreased affinity for ARHGAP12^Set1^ and ARHGAP12^Set2^, and NCK1^Set−1^ and increased affinity for ABL2^Set1^ and BIN1^Set−1^. NCK1^Set−2^ had the same K_d_ value as NCK1^wt^. (**C**) SOS1 co-immunoprecipitated (Co-IP) with NCK1^wt^ but not with NCK1-3^ΔSH3−3^ and NCK1^Set−1^ overexpressed in CHO-K1 cells.

**Figure 4 cells-13-00195-f004:**
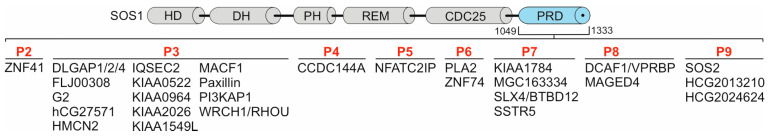
Proteins containing PRMs homologous to P2–P9 derived from the SOS1 PRD. BLAST searches with each SOS1 PRP identified several human proteins (see Appendix A) with a high degree of sequence similarity to P2–P9. CDC25, cell division cycle 25; DH, DBL homology domain; HD, histone-like domain; PH, pleckstrin domain; REM, RAS exchange motif; PRD, proline-rich domain.

## Data Availability

The data supporting the results of this study are available both as Appendix A and, upon reasonable request, are available from the corresponding author.

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
