# Peer review of "Functional Classification and Interaction Selectivity Landscape of the Human SH3 Domain Superfamily"

_cells, 2024, doi:10.3390/cells13020195_

Round 1

Reviewer 1 Report

Comments and Suggestions for Authors

TO AUTHORS

The article by N. S. Kazemein Jasemi et al. addresses important questions related to the binding capacity and affinities of some representative SH3 protein modules. SH3 domain-containing proteins are involved in a variety of biological processes, influencing several diseases. The binding abilities and affinities have been well determined. In conclusion, I think the article can be considered for publication without review.

Author Response

Thank you for reviewing our manuscript. We are grateful for the positive evaluation of our manuscript.

Reviewer 2 Report

Comments and Suggestions for Authors

In this manuscript, Jasemi et.al started from a SH3DCP phylogentic tree and identified clustering pattern to PRM sequences. PRM rich protein SOS1 was selected as a model system to investigate the affinity and selectivity wih FITC and dot blot, and further confirmed with mutant FITC and cell Co-IP. I think in general, the manuscript is well-written, provides new insights to the SH3-PRM binding, and identified new protein protein interaction of SOS1-NCK1 binding. 

I do think the authors should address the following issues:

  1. In Figure S9, the Replicate 2 and Replicate 3 Vinculin band looks identical. Did the author used wrong gel image?
  2. Family #5 and #6 show overlapping PRM sequences with Family #1. What’s the biological interpretation of that?

Other minor issues:

  1. Line 123: It would be better if the author talks more about the conditions of the protease. Also in Table S1, please specify which E.coli strain for the protein expression.
  2. Line 128: 10uM of FITC, line 129: 10ul of glutathione, etc.
  3. Figure S5: in the figure on the left, it should be “PRM”

Author Response

In this manuscript, Jasemi et.al started from an SH3DCP phylogenetic tree and identified clustering patterns in PRM sequences. PRM-rich protein SOS1 was selected as a model system to investigate the affinity and selectivity with FITC and dot blot and further confirmed with mutant FITC and cell Co-IP. I think in general, the manuscript is well-written, provides new insights into the SH3-PRM binding, and identifies new protein-protein interaction of SOS1-NCK1 binding.

Author's Response: Thank you for reviewing our manuscript and providing your valuable comments. We have carefully considered all of your comments, and the incorporation of your suggestions into the revised manuscript has significantly improved its overall quality. Below are our point-by-point responses to each of your comments. The most significant changes in the revised version are highlighted in yellow.

I do think the authors should address the following issues:

  1. In Figure S9, the Replicate 2 and Replicate 3 Vinculin band looks identical. Did the author use the wrong gel image?

Author's Response: Thank you very much for bringing this important issue to our attention. In light of the concerns raised about Figure S9, specifically the identical appearance of the vinculin bands in Replicate 2 and Replicate 3, we identified an error in the gel and promptly replaced it with the correct version. Your attention to detail is appreciated, and the revised figure now accurately represents Figure S9 and the original images.

  1. Family #5 and #6 overlap PRM sequences with Family #1. What’s the biological interpretation of that?

Other minor issues:

  1. Line 123: It would be better if the author talked more about the conditions of the protease. Also in Table S1, please specify which E. coli strain for the protein expression.

Author's Response: Thank you for your suggestion. Text on the cleavage of GST fusion proteins using thrombin has been added on page 3, line 125, and the E. coli strains have been added on page 3, line 121, and Table S1.

  1. Line 128: 10uM of FITC, line 129: 10ul of glutathione, etc.

Author's Response: Corrected.

  1. Figure S5: in the figure on the left, it should be “PRM”

Author's Response: Corrected.

Reviewer 3 Report

Comments and Suggestions for Authors

The manuscript entitled “Functional classification and interaction selectivity landscape 2

of the human SH3 domain superfamily” by Kazemein Jasem et al. and submitted to Cells under manuscript number cells-2796338 presents the functional analysis of shared interacting domains (SH3) of several proteins and focusing specifically on SOS1 as a comprehensive model.

The paper is well written and builds on a previous experience where the authors identified 298 SH3 domains. In this manuscript the authors assess both the phylogenetic and structure-function aspects of these domains with emphasis on the PRM locations. The paper represents a relevant contribution to the understanding of the interaction profile of SH3. Yet, several issues where raised –stated below- and require the authors attention.

One issue to address remains the distinctive/specific interactions for the different PRMs whilst having similar motifs. The specificity of particular biomolecular interactions residing in unique sequences is easily understood and this contrasts strongly with conserved motifs (even more in structurally constrained molecules) that differ only marginally. A partial assessment is done in paragraph 3.4 (indicating precisely the relevance of non-conserved structures) but it would be interesting to have this issue brought forward in more detail and greater extension in this paper and throughout the different disciplines.

115 How was the influence the presence of fluorescein in each of the peptides addressed?

242 What was the rationale to select 25 SH3 domains and not more. What were the representative features?

246 What were the reference peptides and which where the criteria to use them as reference; these reference points are of utmost importance as the standardize the data obtained but may introduce a common bias that is neglected.

256 Why where binding experiments not done with realtime monitoring techniques that do not require labeling (and hence potential interference in the binding) such as ITC, SPR of SAW?

How does the family classification affect the individual assessment of a particular SH3 domain, in other words what is the biological value of such classification; can any impact from such classification be routed to the spectrum of diseases where SH3DCPs allegedly play a role and hence, is there any clinical assessment associated? This is marginally addressed in the last part of the discussion but should be the broader focus of scientific research without ending in rather empty phrases the include “could be of importance” or “may have far-reaching implications”.

The authors are encouraged to address the minor issues as well as to give the broader perspective of their findings some thoughts and the manuscript could be recommended for publication in Cells.

Author Response

Reviewer #3:

The manuscript entitled “Functional classification and interaction selectivity landscape of the human SH3 domain superfamily” by Kazemein Jasem et al. and submitted to Cells under manuscript number cells-2796338 presents the functional analysis of shared interacting domains (SH3) of several proteins and focuses specifically on SOS1 as a comprehensive model. The paper is well written and builds on a previous experience where the authors identified 298 SH3 domains. In this manuscript, the authors assess both the phylogenetic and structure-function aspects of these domains with emphasis on the PRM locations. The paper represents a relevant contribution to the understanding of the interaction profile of SH3. Yet, several issues were raised –stated below- and require the author’s attention.

Author's Response: Thank you for reviewing our manuscript and providing your valuable comments and constructive suggestions. We have carefully considered all of your comments, and the incorporation of your suggestions into the revised manuscript has significantly improved its overall quality. Below are our point-by-point responses to each of your comments. The most significant changes in the revised version are highlighted in yellow.

One issue to address remains the distinctive/specific interactions for the different PRMs whilst having similar motifs. The specificity of particular biomolecular interactions residing in unique sequences is easily understood and this contrasts strongly with conserved motifs (even more in structurally constrained molecules) that differ only marginally. A partial assessment is done in paragraph 3.4 (indicating precisely the relevance of non-conserved structures) but it would be interesting to have this issue brought forward in more detail and greater extension in this paper and throughout the different disciplines.

Author's Response: We appreciate your insightful comment on the unique interactions of PRMs with similar motifs. Your suggestion to further explore this aspect across disciplines is well taken. In the revised manuscript (page 10, line 391), we have provided a more thorough description of the specificity of variable residues within conserved PRMs, emphasizing their critical role in defining the specificity of the interaction with SH3 domains.

115 How was the influence of the presence of fluorescein in each of the peptides addressed?

Author's Response: We appreciate you raising this concern. We generated only fluorescein-labeled peptides in this study and did not perform a comparison with unlabeled peptides. However, our observations in the present study suggest that although we observe specific interactions with SH3 domains for many peptides, not all peptides interact with each SH3 domain. Furthermore, in a previous study (Ref. 22), we have shown that fluorescein labeling has no effect on the interaction of a proline-rich peptide (e.g., RP1) with an SH3 domain (e.g., GRB2-1) using fluorescence polarization with fluorescein-labeled peptide and isothermal titration calorimetry (ITC) with label-free peptide. In addition, mixing fluorescein-labeled peptides with SH3 domains in a stopped-flow instrument did not change the fluorescence signal (Ref. 32), suggesting that the presence of fluorescein does not significantly affect the interaction of PRMs with SH3 domains. We revised the following text on page 6, line 271: “In a previous study [22], we have shown that fluorescein labeling does not affect the interaction of proline-rich peptides with the SH3 domains.”

242 What was the rationale for selecting 25 SH3 domains and not more? What were the representative features?

Author's Response: The rationale behind the selection of SH3 domains in our study was based on ensuring a representative sample while considering computational, structural, and experimental feasibility (see Tables S3 and S4). We aimed to select SH3 domains from different families with analyzed PRM interactions, prioritizing key proteins in signaling pathways that are experimentally feasible to purify and exhibit stability. Despite challenges in purifying all selected SH3 proteins, we successfully purified 25 proteins and measured their interactions. In addition, we ensured that we had at least one representative SH3 domain per defined family. We revised the following text on page 6, line 260: “We selected at least one representative SH3 domain per defined family concerning accessibility and experimental viability (see Tables S3 and S4).”

246 What were the reference peptides and what were the criteria for using them as a reference; these reference points are of utmost importance as they standardize the data obtained but may introduce a common bias that is neglected.

Author's Response: Thank you for raising this important issue. The reference peptides used in this study include reference peptide 1, derived from SOS1, and reference peptide 2, derived from WRCH1/RHOU, as shown in Table S2. These two peptides were chosen as references because of their previously measured high-affinity interactions with SH3 domains in various studies. Although our primary focus in this study was on the PRMs of the SOS1 protein model, we decided to include a reference peptide from another protein and thus used the second reference peptide from WRCH1/RHOU for comparison purposes. The use of reference peptides should indeed be a means of standardization of the data obtained, but it is only partially successful (Table S6). Therefore, we have added the following sentence in the revised manuscript on page 2, line 90: “The reference peptides RP1, a derivative of SOS1 and part of P3, and RP2, a derivative of the RHO GTPase WRCH1, were used as controls.”

256 Why were binding experiments not done with real-time monitoring techniques that do not require labelings (and hence potential interference in the binding) such as ITC, SPR, or SAW?

Author's Response: Thanks for pointing this out. We have both ITC and SPR from the proposed instruments and have experience with ITC in measuring SH3-PRP interactions (Ref. 22). However, we would need 10 times more proteins and 50 times more peptide for each ITC measurement, which provides equilibrium Kd values as well as fluorescence polarization; for the latter, we used a quartz glass fluorescence cuvette (Hellma Ultra-Micro Cuvette 105.250-QS) with a reaction volume of 200 µl. Using the SPR instrument requires immobilization of the peptide and titration of the SH3 domain, so the peptide must be labeled with, for example, a His tag, which would have been beyond our budget. However, the advantage of the FITC-labeled peptides for our study was the combination of both dot blotting and polarization methods. Possible interference of the SH3 domain-PRP interaction can be excluded because we did not observe any change in the fluorescence signal when mixing FITC-labeled peptides with SH3 domains in a stopped-flow instrument (Ref. 32). We revised the following text on page 3, line 141: “The interaction between fluorescein-labeled proline-rich peptides (0.2 µM) and increasing concentrations of SH3 domains (ranging from 0 to 200 µM) was measured in a buffer, containing 30 mM Tris/HCl (pH 7.5), 5 mM MgCl2, and 3 mM DTT at 25°C, and a total volume of 200 µl using a Fluoromax 4 fluorimeter in polarization mode and a quartz glass fluorescence cuvette (Hellma Ultra-Micro Cuvette 105.250-QS, Thermo Fisher Scientific).”

How does the family classification affect the individual assessment of a particular SH3 domain, in other words, what is the biological value of such classification; can any impact from such classification be routed to the spectrum of diseases where SH3DCPs allegedly play a role and hence, is there any clinical assessment associated? This is marginally addressed in the last part of the discussion but should be the broader focus of scientific research without ending in rather empty phrases that include “could be of importance” or “may have far-reaching implications”.

Author's Response: Thank you very much for bringing this important issue to our attention. We have focused primarily on the sequence-structure-function relationships of the SH3 domain-proline-rich motif interaction, rather than on the biological value and possible clinical evaluation associated with the SH3 domain classification. Therefore, we have begun to correlate our results with the possible involvement of SH3 domains in disease. Our concept was to delineate the association of each family with a specific disease or to identify the involvement of specific interface residues in specific diseases. However, there is limited information available in the literature on this aspect. Therefore, we have omitted the following text from the Discussion section (Page 13, line 540): “Using bioinformatics and phylogenetic trees, complemented by biophysical measurements, our goal is to decipher the intricate matrix that governs these interactions. This predictive approach may have far-reaching implications for the understanding of cellular signaling pathways and the development of novel therapeutic interventions.”

While our study provides a comprehensive analysis of the interactions and new biophysical insights into SH3-PRMs interactions for each family, the assessment of the biological value of such classification fell short. Therefore, we have included a more comprehensive discussion and a new "Conclusions" section in the revised manuscript, which now provides a broader perspective on our findings.

The authors are encouraged to address the minor issues as well as to give a broader perspective of their findings and thoughts and the manuscript could be recommended for publication in Cells.

Author's Response: We appreciate your encouraging comments and constructive feedback. We have carefully addressed the minor issues raised. A more comprehensive discussion and conclusions of our findings are now included in the revised manuscript. We thank you for your consideration and are confident that we have improved the manuscript to meet the standards for publication in Cells.